# Ventilation Difficulty Caused by Obstructed Heated Breathing Circuit

**DOI:** 10.3390/medicina59050965

**Published:** 2023-05-17

**Authors:** Jiyeon Kwon, Eunsu Kang, Sunghyun Shin, Byeongcheol Lee, Myoungjin Ko, Sehun Kim, Soojee Lee

**Affiliations:** 1Department of Anesthesia and Pain Medicine, Inje University Haeundae Paik Hospital, 875 Haeundae-ro, Haeundae-gu, Busan 48108, Republic of Korea; 2Department of Anesthesia and Pain Medicine, Inje University Busan Paik Hospital, 75 Bokji-ro, Busanjin-gu, Busan 47392, Republic of Korea

**Keywords:** ventilation, hypothermia, equipment failures, mechanical ventilator

## Abstract

For perioperative hypothermia prevention, a heated, humidified breathing circuit equipped with a fluid-warming unit inside the inspiratory limb has been developed. We report a ventilation difficulty caused by an obstructed heated breathing circuit. Cotton surrounding the hot wire, temperature sensor, and fluid tubing in the distal inspiratory limb was irregularly thicker than that of a normal circuit and nearly blocked the lumen. Despite carrying out routine checks on the anesthesia workstation preoperatively, we failed to make a prediagnosis by omitting the flow test after changing the circuit. This case puts emphasis on a routine flow test with a meticulous examination of the heated breathing circuit before every procedure.

## 1. Introduction

Many studies have advocated for the benefits of perioperative hypothermia prevention. This condition increases not only the risk of blood loss and transfusion [1], adverse myocardial events [2], cancer recurrence [3], and wound infection but also hospitalization time [4]. Several measures have been established to maintain normothermia in anesthetized patients. These include preoperative warming, cutaneous warming using forced air or heated water blankets, intravenous fluid warming, and heated as well as humidified gas inspiration. Using a heated, humidified breathing circuit provides additional advantages by improving mucus transportability [5], which reduces the severity of postoperative sore throat and cough [6]. Recently, a heated, humidified breathing circuit equipped with a fluid-warming unit inside the inspiratory limb was developed. The dual effect of both these techniques can effectively help maintain a patient’s intraoperative core temperature [7]. However, heating accessories inside breathing circuits can place patients at risk. For instance, case studies have reported fires inside breathing circuits caused by a heating system electrical shorting [8] and contact burns from a heated breathing circuit [9]. Here, we report, for the first time, a rare case of ventilation difficulty caused by an obstructed heated breathing circuit.

## 2. Case Report

A 54-year-old man was scheduled for the removal of an L2–3 epidural tumor that was the suspected cause of radiating pain in his right leg. He had underlying atrial fibrillation and coronary artery occlusive disease. Recent transthoracic echocardiography revealed the preserved left ventricular systolic function. On the day of the scheduled surgery, a ventilator (Aestiva/5, Datex-Ohmeda, Inc., Madison, WI, USA) checkup was performed at the start of the day shift using a conventional breathing tube. For effective temperature maintenance throughout the 5 h long surgery, we used a heated, humidified breathing circuit (Mega Acer Kit, Ace Healthcare, Okcheon-gun, South Korea). After preparing the circuit, the patient was preoxygenated and general anesthesia was induced using remifentanil and propofol. We employed a target-controlled infusion model for total intravenous anesthesia. As the patient lost consciousness, mask ventilation with 100% oxygen was commenced. However, ventilation difficulty was encountered. Adjusting the pressure-limiting valve to 40 mmHg yielded a tidal volume of <100 mL, with nearly undetectable end-tidal CO_2_. A two-handed technique using an oropharyngeal airway and rocuronium administration did not improve ventilation. Preoperative evaluation determined that the patient had a Mallampati score of 2 and body mass index of 26 kg/m^2^. He had no history of obstructive sleep apnea, and he was neither edentulous nor bearded. Therefore, the patient was not considered to be at a high risk of difficult ventilation. To rule out ventilator-associated causes, a self-inflating bag was used instead. Ventilation improved significantly. We concluded that the ventilator was faulty, but when the circuit was connected to a new ventilator, the difficulty persisted. This was resolved after exchanging the circuit. The careful examination of the original circuit revealed that the cotton surrounding the hot wire, temperature sensor, and fluid tubing in the distal inspiratory limb was irregularly thick in comparison to that of a normal circuit and was nearly blocking the lumen (Figure 1). The patient was stable throughout the event and surgery. He was uneventfully discharged from the post-anesthesia care unit.

## 3. Discussion

To our knowledge, this is the first case of an obstructed heated breathing circuit. While not all heated breathing circuits contain a device within the tubing, in this case, the lumen of the heated breathing circuit is partially occupied by its distinctive inner device. Figure 2 shows that fluid tubing and accessory tubing, along with the temperature sensor and hot wire, are covered with cotton and placed inside the inspiratory limb of the circuit. Only a slight increase in the circumference of the inner device can impede the inspiratory flow through the circuit. In our case, we found that an irregularly thick cotton layer had almost completely blocked the lumen, causing unexpected ventilation difficulty. We immediately reported this finding to the manufacturer, who confirmed it was a manufacturing error. Since then, the manufacturer has reinforced their quality control and we have not encountered any similar issues to date.

Obstructions in conventional breathing circuits have been described before. Yang et al. [10] reported a case of an inspiratory circuit limb that was partially sealed with excessive adhesive. Preoperative self-testing and leakage testing did not detect this obstruction. The obstruction caused mask ventilation difficulty mimicking severe bronchospasm. In addition, Eckhout and Bhatia [11] presented a case of expiratory limb occlusion in an extended circuit that caused ventilation difficulty. This complete occlusion was not observed because the testing of the circuit was performed before the addition of circuit extensions. Luckily, in both cases, patients were stable throughout the event.

The cause of unexpected difficult ventilation can be hard to pinpoint. Without prompt intervention, it may lead to hypoxia, brain damage, or death. Different mechanisms contribute to difficult ventilation. Age > 55 years, obesity, the presence of a beard, lack of teeth, and a history of snoring are known patient-related risk factors for difficult mask ventilation [12]. The operator’s skill in gaining a tight seal while securing the pharyngeal airway is also important. Above all, well-functioning equipment is the basis for safely ventilating a patient. This includes the confirmation of an appropriately sized mask and airway, an intact breathing circuit, and a functioning valve. Mehta et al. [13] analyzed the American Society of Anesthesiologists (ASA) closed claims data related to gas delivery equipment from the 1970s to the 2000s. Forty claims were reported between 1990 and 2011. Among them, eight (20%) cases reported patient injuries caused by a breathing circuit problem. Similarly to our case, five (63%) breathing circuit claims involved a provider error with equipment failure. Notably, six (75%) breathing circuit claims were deemed preventable by performing a pre-anesthesia machine check.

The ASA 2008 pre-anesthesia checkout procedure guidelines list eight items to be checked prior to every procedure [14]. This includes verifying unobstructed gas flow through the breathing circuit by manually using an extra breathing bag as a test lung or using automated check routines. Although most modern anesthesia workstations provide automated check routines, according to Dosch [15], their accuracy varies. Adequate flow should be established using both manual and automated checks. Moreover, similarly to the leakage test, this flow test should be performed using the circuit that will be used during anesthesia.

In our case, the ventilator we used did not provide an automated check routine so manual checkout was performed preoperatively. Although the complete check was carried out, we failed to detect the obstruction by omitting the flow test using the specific circuit when changing the circuit type just before induction. Our case report has a limitation, in that it lacks more detailed data regarding airway pressure, end-tidal CO_2_, or chest auscultation, which would have provided a clearer understanding of the situation.

## 4. Conclusions

While a heated, humidified breathing circuit offers several advantages during anesthesia, it also increases the risk of obstruction. Therefore, conducting a routine flow test adhering to guidelines and meticulous examination before every procedure is essential. Additionally, in cases of unexpectedly difficult ventilation, gas delivery equipment failure should always be considered as a potential cause.

## Figures and Tables

**Figure 1 medicina-59-00965-f001:**
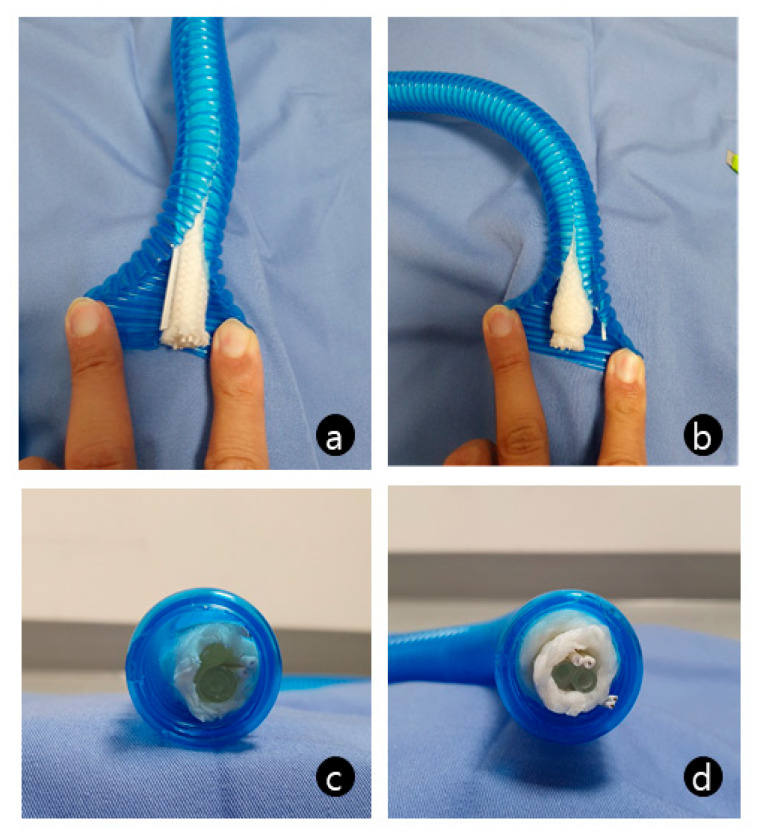
Comparison of a normal heated breathing circuit (**a**,**c**) with an obstructed circuit (**b**,**d**).

**Figure 2 medicina-59-00965-f002:**
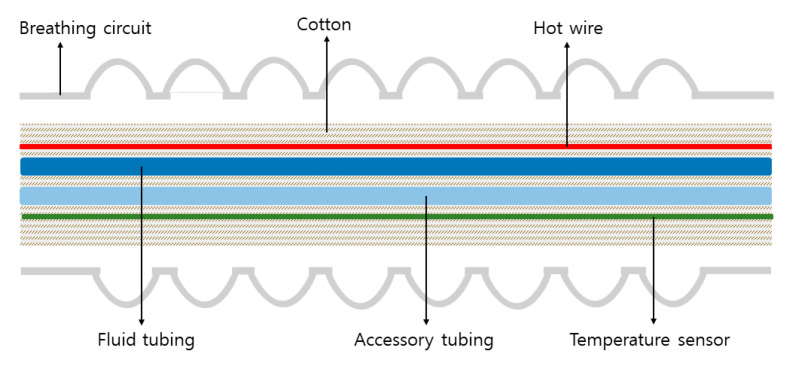
Schematic diagram of a heated breathing circuit.

## Data Availability

The data of this study are available from the corresponding author upon reasonable request.

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
