# Peer review of "Ventilation Difficulty Caused by Obstructed Heated Breathing Circuit"

_medicina, 2023, doi:10.3390/medicina59050965_

Round 1

Reviewer 1 Report

The authors of this article have aimed at presenting a rare case of ventilation difficulty caused by an obstructed heated breathing circuit. I have some considerations:

1. The abstract should contain no subtitles (please revise the journal's submission guidelines). Also, I do not feel the flow of text in the abstract. It should be more logical, beginning with the background, then the aim, followed by the case presentation, any valuable discussions about the case, and finally, a conclusion that puts this into the context of practice. 

2. Please replace the subtitle "Results" with "Discussion".

3. Please spell out every number that starts a sentence. Some examples are in lines 87 (40 claims), 88 (8 cases), and 89 (5 breathing circuits). Please check for similar errors. 

4. Line 25: The authors used "not only" but was not followed by "but also".

5. Line 28: Wrong use of the semicolon. Please use commas instead.

6. Line 29: Please replace the comma with "as well as".

7. Formatting error: Some paragraphs are justified and others are not. Please justify all paragraphs for the sake of consistency.

8. Please state the limitations of this case report. After all, according to your statements, this was only reported in your setting. Can you please explain why this happened in the first place?

9. Please also discuss whether this has something to do with the manufacturer, delivery, or storage. 

10. Please discuss whether your hospital conducted a root cause analysis. If not, this is a huge limitation as we are unable to detect the cause of the problem, and now only the doctor and/or nurse anesthetist are to blame. 

11. Please consider adding a conclusion to discuss further implications of this case and generalize this to a wider audience.

12. Please revise the manuscript for any further English-language proofreading errors. I recommend using a controlled AI like Grammarly or Microsoft Editor. It would also be helpful to have a couple of native eyes go through the manuscript to identify any slight errors.

Reviewer 2 Report

The authors report a case in whom restricted air flow was found during induction, mimicking bronchospasm. Changing the ventilator did not result in an improved situation nor administration of curarisation; checking meticulously the breathing circuit revealed the inspiuatory limb of the breating circuit was nearly completely occluded by a thickenend cotton wool; resulting a thickenend distal inspiratory limb.

The case is well described and the report well written.

Comments

The authors report gas flow is well reduced. It would be interesting to see some more data. Also, interesting would be a the capnogram during manual ventilation. This should be provide already the clue of what was happening with the inspiratory flow.

Not all heated breathing circuits use cotton wool inside the tubing. Somtimes, only a heating wire is used besides a sensor at the end of the tubing, significantly reducing the risk of obstruction. Alternatives on the market should be taken up in the discussions’ section.

As the authors remark correctly, automatic checking of the flows before start of the induction, would have revealed ths issue. The authors should state which aaesthesia machine was used and whether such a flow check was possible before start of anesthesia.

Round 2

Reviewer 1 Report

Thanks for incorporating the changes. All's fine now. Good luck.

Author Response

Dear Reviewer 1.

Thank you for your time and effort to comment on our work. Your comments were very valuable and I believe these would make our manuscript higher quality. 

Reviewer 2 Report

The authors revised and improved considerably the manuscript following the. guidance of the reviewers. They admitted that some limitations are present such as absence of a capnogram, which should have provided the cause of the air flow obstruction immediately. The addition of a capnogram would augment the quality and the educational value of the manuscript considerably. The authors could add a designed capnogram as they perceived it during induction of anesthesia?
